# From months to minutes: Creating Hyperion, a novel data management system expediting data insights for oncology research and patient care

**Eric Snyder, Thomas Rivers, Lisa Smith, Scott Paoni, Scott Cunliffe, Arpan Patel, Erika Ramsdale** *

James P. Wilmot Cancer Institute, University of Rochester Medical Center, New York, United States of America

* Erika_ramsdale@urmc.rochester.edu

**Data Availability Statement:** All data are in the manuscript and/or supporting information files.

## Abstract

Here we describe the design and implementation of a novel data management platform for an academic cancer center which meets the needs of multiple stakeholders. A small, cross-functional technical team identified key challenges to creating a broad data management and access software solution: lowering the technical skill floor, reducing cost, enhancing user autonomy, optimizing data governance, and reimagining technical team structures in academia. The Hyperion data management platform was designed to meet these challenges in addition to usual considerations of data quality, security, access, stability, and scalability. Implemented between May 2019 and December 2020 at the Wilmot Cancer Institute, Hyperion includes a sophisticated custom validation and interface engine to process data from multiple sources, storing it in a database. Graphical user interfaces and custom wizards permit users to directly interact with data across operational, clinical, research, and administrative contexts. The use of multi-threaded processing, open-source programming languages, and automated system tasks (normally requiring technical expertise) minimizes costs. An integrated ticketing system and active stakeholder committee support data governance and project management. A co-directed, cross-functional team with flattened hierarchy and integration of industry software management practices enhances problem solving and responsiveness to user needs. Access to validated, organized, and current data is critical to the functioning of multiple domains in medicine. Although there are downsides to developing in-house customized software, we describe a successful implementation of custom data management software in an academic cancer center.

## Author summary

Ensuring timely access to accurate data is critical for the functioning of a cancer center. Despite overlapping data needs, data are often fragmented and sequestered across multiple systems (such as the electronic health record, state and federal registries, and research

**Funding:** ER is supported by the National Cancer Institute (K08CA248721) and the National Institute on Aging (R03AG067977). The funders had no role in study design, data collection and analysis, decision to publish, or preparation of the manuscript.

**Competing interests:** The authors have declared that no competing interests exist.

databases), creating high barriers to data access for clinicians, researchers, administrators, quality officers, and patients. The creation of integrated data systems also faces technical, leadership, cost, and human resource barriers, among others. The University of Rochester Wilmot Cancer Institute (WCI) hired a small team of individuals with both technical and clinical expertise to develop a custom data management software platform addressing five challenges: lowering the skill level required to maintain the system, reducing costs, allowing users to access data autonomously, optimizing data security and utilization, and shifting technological team structure to encourage rapid innovation. We describe how this platform, Hyperion, was successfully designed, developed, and implemented at WCI. We offer an overview of the data architecture, provide insight into the design elements that address our identified challenges, and discuss the performance of the system in terms of cost, speed, and user engagement.

## Background and significance

Academic cancer centers, particularly those embracing the Learning Healthcare Systems (LHS) model to dynamically generate and utilize high-quality evidence for patient decision making [1], require integration and maintenance of data systems offering intuitive access and manipulation of valid, ordered, and up-to-date knowledge informing clinical operations, clinical decision support, and research. Electronic health record (EHR) systems do not offer the data curation nor the user experience required to fully meet these needs. EHR systems were primarily designed to improve billing and revenue capture, requiring very different design decisions which often result in clunky, burdensome, and disorganized systems from the perspectives of many end-users. Moreover, useful data are not exclusively stored in a single location like the EHR, but across dozens of databases utilizing disparate (and often incompatible) technologies. Addressing the data needs of an academic cancer center introduces many challenges, including recruitment and retention of the appropriate technical expertise, while adhering to already thin financial budgets. [2] Technical difficulties include seamless integration of multiple data sources, enhancement of user buy-in for the data system (including mitigation of technology burnout), and rapidly changing technical and data landscapes. [2] Leadership challenges implicate the dominant paradigm for vertical, clinician-centric decision-making: current organizational leadership structures may be ill-suited to devising technical data solutions requiring systems thinking and rapid adaptation/iteration. Importing organizational processes, systems thinking approaches, and technical domain expertise from other industries could help academic cancer centers around the country surmount many issues impeding data utilization.

Attempts to optimally balance data currency, access, validation, and integrity in the healthcare typically involve data or research warehouses. [3,4] Given the disparate nature of the data sets to be merged, as well as the heightened security and privacy concerns involved in storing patient data, barriers include large capital outlays and accommodating the potentially competing design aims such as efficiency, timeliness of reports, user experience, data validity, and accuracy. [5] Different health systems, or even different groups within an individual health system, may prioritize different design aims, such that a one-size-fits-all technical solution is inadvisable and obviates the ability to purchase a ready-made software solution. Even if a ready-made solution is available, the shifting data landscape could quickly make it obsolete; maintenance of data architecture, not only its initial build and deployment, requires significant ongoing technical skill and time. Throughout the processes of data architecture design and

implementation, ongoing user feedback is critical, and clinical domain expertise is required at every step to maximize utility, comprehensiveness, and validity.

This paper discusses a novel design and successful implementation of a sophisticated data architecture to address the data needs of an academic cancer center. Although each center has individualized needs embedded in idiosyncratic circumstances, a few principles may be derived to guide other centers hoping to implement and maintain a customized data architecture that users can employ confidently, productively, and adaptively to facilitate rapid answers to quality and research questions and ultimately to improve patient outcomes at the point of care.

## Materials and methods

### Wilmot cancer institute

The James P. Wilmot Cancer Institute (WCI) at the University of Rochester Medical Center (URMC, a large tertiary care academic medical center) is the largest cancer care provider in upstate New York, with a catchment area of 3.4 million people across a 27-county region. Prior to mid-2019, patient and research data were distributed across multiple unconnected systems, including the EHR (Epic), Research Electronic Data Capture (REDCap) [6] databases, clinical trial management software, individually maintained disease databases, laboratory information management systems, shared resource databases, and billing applications. Additionally, useful data outside of the institution existed in a variety of formats, including behind web portals (e.g., clinicaltrials.gov), [7] in private company databases (e.g., externally performed molecular tumor testing and nonprofit organization public health databases), and within various externally maintained registries (e.g., New York State cancer registry). Gathering, merging, and validating data were tedious, time- and resource-intensive procedures performed largely manually, resulting in high expenditures for human abstractors and significant delays in implementing data insights. Data users lacked the ability to interact with real-time data; while static reports could be generated, the reliance on manual effort led to outdated, delayed reports. Intuitive and interactive data visualization was unavailable, limiting data exploration necessary to develop a research question or protocol, review clinical data, probe quality issues, or refine operations.

In 2019, a small informatics team was assembled to address WCI's data challenges, consisting of two software architects with expertise in custom healthcare software, a business intelligence software developer, a project manager, and a clinician with expertise in data science and quantitative research methods. The initial primary aim was to improve data availability to WCI faculty and staff, but in collaboration with the data architects it was clear that other aims should be equally prioritized, including data security, access to near-real-time data, data validity, flexibility in integrating data sources, and a platform for interacting with and visualizing data. Between May 2019 and December 2020, a complete data management and analytics platform was built, iterated, and deployed for WCI users: the Hyperion Centralized Medical Analytics (Hyperion) platform. The design and development process was conducted using Project Management Body of Knowledge (PMBOK) best practices under the guidance of the project manager, including scoping, planning, identifying and meeting with stakeholders for each design phase, and communicating requested changes to the development team. [8]

### Medical informatics challenges

Although well-maintained data warehouses are critical to the functioning of academic institutions, they do not address all informatics needs. All data management solutions share the common goal of consolidating data and ensuring its quality (ensuring, for example, data validity,

availability, and completeness). In addition to meeting this goal for WCI, we sought to overcome five main challenges with the implementation of Hyperion. These challenges, abstracted from our iterative design process including input from all stakeholders, are distilled principles which could be considered across diverse implementations rather than a comprehensive description of discrete implementation challenges we faced.

### Challenge 1: Lowering the skill floor

Setting up a data warehouse and maintaining complex interfaces, dashboards, and ad-hoc reporting often require significant time and large teams of information technology professionals. In one instance, developing a single-purpose, straightforward data management system to study antimicrobial resistance required two years and 4000 person-hours among four skilled personnel. [9] The personnel costs of creating such systems become the primary challenge in implementation. [10] Beyond deployment, data management systems must respond to constantly updating data sets and sources, as well as updated user needs; for optimal functioning, maintenance typically requires continued involvement of highly educated (and high-cost) professionals such as software developers and data architects to design changes, as well as a team of programmers to directly implement changes to the software code. Budgets for these activities can quickly become unsustainable. Resilient software design, automation of projected maintenance activities, and creation of interfaces to translate programming and auditing activities could potentially "lower the skill floor," allowing a smaller team of non-experts to substitute for many of the activities of much larger (and more costly) information technology teams.

### Challenge 2: Reducing the cost of technology

The technical architecture to achieve data storage and processing can be very expensive, whether it exists on-premise or in cloud-based systems. On-premise solutions have high start-up costs for processing and storage hardware. Operating a cloud-based system similar in size to Hyperion would be expected to cost about $35,000 per month, using Amazon Web Services as a benchmark. [11]

### Challenge 3: Enhancing user autonomy

Data users exist on a spectrum of familiarity and sophistication with data manipulation, ranging from users who want simple, intuitive visual summaries to those capable of sophisticated analysis of raw data. A data system needs to be accessible and usable across the spectrum of potential users within WCI, ideally without the need for manual creation of curated datasets and visuals. Users should be able to autonomously access the data and tools they need to answer their own queries as much as possible.

### Challenge 4: Optimizing data governance

Patient data are subject to laws and policies (such as provisions within the Health Insurance Portability and Accountability Act [HIPAA]) enforcing high scrutiny and standards to maintain patient privacy rights. Integration of data governance policies and activities into the data management platform would facilitate adherence to the strict confidentiality guidelines of the health system. Additionally, a well-designed ticketing system can help users clarify their requests, ensure appropriate data usage, streamline ticket approval and completion, and permit ongoing user needs assessment.

### Challenge 5: Changing technological team structure

The culture of academic medical centers often promotes top-down decision-making, prioritizing a particular paradigm centered around clinicians. Although clinicians bring invaluable perspective to the design and usage of data systems, they are not typically trained in (or necessarily aware of) the specific technical skillset required to design sophisticated data management systems. Leadership structures with clinicians or administrators at the "top" can hinder technical teams, limiting their autonomy to implement technical best practice decisions, and delaying even simple architectural and programming tasks. Flattening the decisional hierarchy and implementing a transdisciplinary team approach could optimize functioning and increase development speed, simulating the pace of industry teams.

## Results

The custom-built data management solution for WCI, Hyperion, consists of a back-end relational Structured Query Language (SQL) database linked to a front-end interface platform accommodating multiple user types and tasks, including database administration, ticketing, reporting, and user dashboards. Table 1 compares Hyperion features to those of the most commonly available clinical data management software packages. Table 2 summarizes how Hyperion design addresses the 5 identified challenges above.

**Table 1. Comparison of features: Hyperion versus other commercially available, commonly used clinical data management software systems.**

| Feature | Hyperion | Most-Used CDMs* |
|---|---|---|
| Custom Security | X | X |
| User Defined Roles | X | X |
| Security on all Individual Data Elements | X | |
| Dozens of Built In Integrations (EHR/CTMS/XML/State APIS) | X | |
| Clinical Trial Data Integrated | X | |
| Clinical Operational Data Integrated | X | |
| Outside Vendor Data Integration | X | |
| Built in Analytics Packages | X | X |
| Ability to Use Outside Analytics (Tableau, QLIK, etc) | X | X |
| Ability to Embed Outside Analytics (Tableau, QLIK, etc) | X | |
| Built in Learning Management Systems | X | |
| Built in Data Governance Systems | X | |
| Built in Document Management Systems | X | |
| Automated Emailing of Reports | X | X |
| Federated Data | X | X |
| Plug and Play Custom Application Support (Write your own app) | X | |
| Change Management Integration | X | |
| Full System Auditing | X | X |
| Social Determinates of Health Integration | X | |
| Built in Ticketing System | X | |
| GeoSpatial Analysis Tools | X | |
| Web Accessible/Mobile Accessible | X | X |
| Custom Data Fields | X | X |

*CDM = Clinical Data Management software

**Table 2. Design elements supporting identified challenges.**

| Challenge | Design Element |
|---|---|
| *Lowering Skill Floor* | Configuring a new data source for import is automated via a point-and-click graphical user interface (GUI) |
| | Custom wizards embedded in GUI walk administrators through setup of data interfaces, new user profiles, and access privileges |
| | Automatic parsing of imported database schema to inform wizard setup |
| | Custom code to facilitate import, automatic renaming, and validation of EHR data |
| | Automated dashboards for visually monitoring data validity and metrics |
| | Custom wizard walks users through resolving data validation issues identified by HDM |
| *Reducing Technology Cost* | Use of multi-threaded processing |
| | Use of open-source programming languages for all code |
| | Automatically matches addresses to national database using a "certainty percentage", allowing users to specify a threshold and limit manual review |
| *Enhancing User Autonomy* | Secure sandboxing with curated datasets |
| | Customizable dashboards (see Table 3) |
| | Custom geospatial software supporting user-specified map overlays (CANVAS) |
| | User tracking of submitted tickets enabled in Hyperion |
| *Optimizing Data Governance* | Centralized data governance model with decentralized execution by users (e.g. in sandboxes or dashboards) where feasible |
| | All requests linked to user-generated tickets |
| | Data governance policy embedded in Hyperion, which hard-stops review and signatures by users when updated, and stores documents with signatures |
| | Robust Data Governance Committee which meets monthly and includes all key stakeholders |
| | Pre-specified criteria and processes for approval, scoping, and management of tickets |
| *Changing Team Structure* | Informatics team operates internally with flattened decisional hierarchy |
| | Transdisciplinary team including technical, management, and clinical expertise |
| | Formal authority for interaction with other institutional structures resides within a co-directed role (one technical and one clinical lead) |
| | Recruitment and hiring processes value and emphasize diversity of identity, perspective, background, and training/skillset |

## Hyperion data manager

Hyperion's "nervous system" is the Hyperion Data Manager (HDM), an interface and validation engine built utilizing the Python programming language. Via a front-end system utilizing a Flask-based graphical user interface (GUI), approved users access point-and-click tasks to import data sources, initiate and manage extract, transform, and load (ETL) procedures, create sandboxes, copy table structures, re-run interfaces, and check for errors. A built-in custom wizard permits users to set up data interfaces without any programming knowledge for most data sources; bidirectional interfaces can be easily configured for all common data formats and models (HL7 FHIR, XML, API etc.).

For integrating any new data source, HDM reads the database schema for the new data source and presents approved administrative users with a field listing. Via the GUI, users can select what to import into the Hyperion platform, create table names, rename data elements, and select a data update interval. HDM creates a scheduled interface pull at the selected interval and can accommodate near-real-time intervals. HDM can read in the most utilized database technologies such as Microsoft's SQL Server and Oracle, and it also supports token authentication, which is utilized by multiple governmental data sources and common medical database software products such as REDCap. For EHR data, HDM has custom code to simplify

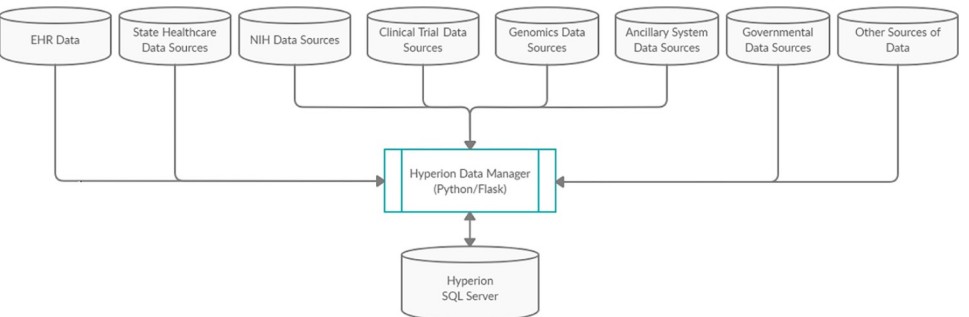

**Fig 1. Current architecture of Wilmot Cancer Institute (WCI) data warehouse.**

and semantically define tables and interpret names for commonly used fields It translates field names into more readable form based on clinical naming conventions and uploads data directly at the specified intervals. This streamlines table design and reduces time delay for downstream needs including ad-hoc report development, sandbox creation, and full-scale applications. As all data tables are normalized and explicitly defined within our system, data reusability across the system is also facilitated.

Ease of use combined with robust procedures to ensure data quality has permitted rapid integration of multiple data sources for a variety of clinical, quality, operational, and research purposes (Fig 1). To increase oversight of data validity and currency, HDM incorporates a complete validation and metrics monitoring package for data uploaded to Hyperion. HDM stores and displays metrics including number of new records per interface, timing and number of updates, and deletions or modifications to records since the prior interface update. Automated dashboards allow non-expert users to monitor metrics (Fig 2); the dashboard displays are interactive, allowing point-and-click activities (e.g., adding or removing metrics to the display) and hovering over data points to get more information (see Figs A-C in S1 Text for further examples).

HDM's validation routines regularly evaluate for data consistency issues such as field and data type mismatches and extreme shifts in table sizes indicating mass data deletion or

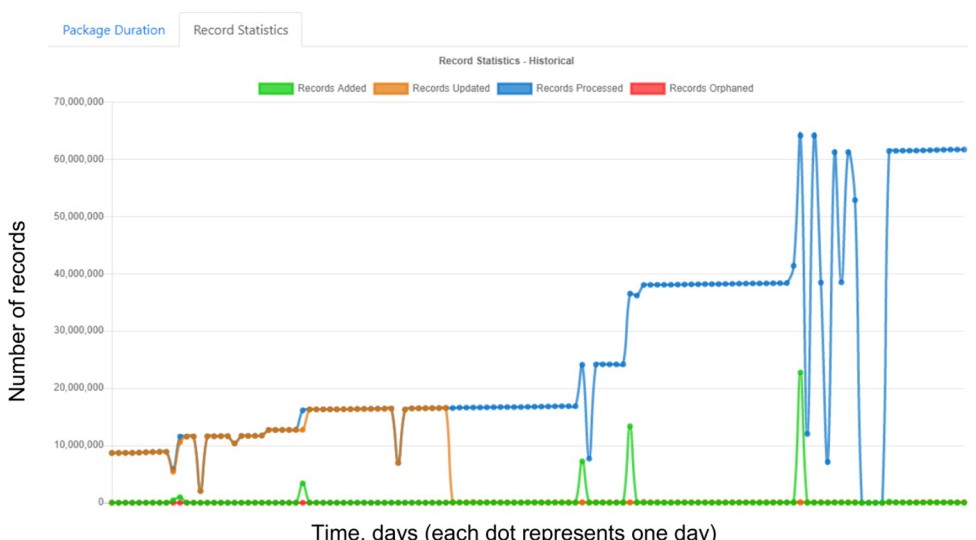

**Fig 2. HDM monitoring dashboard.**

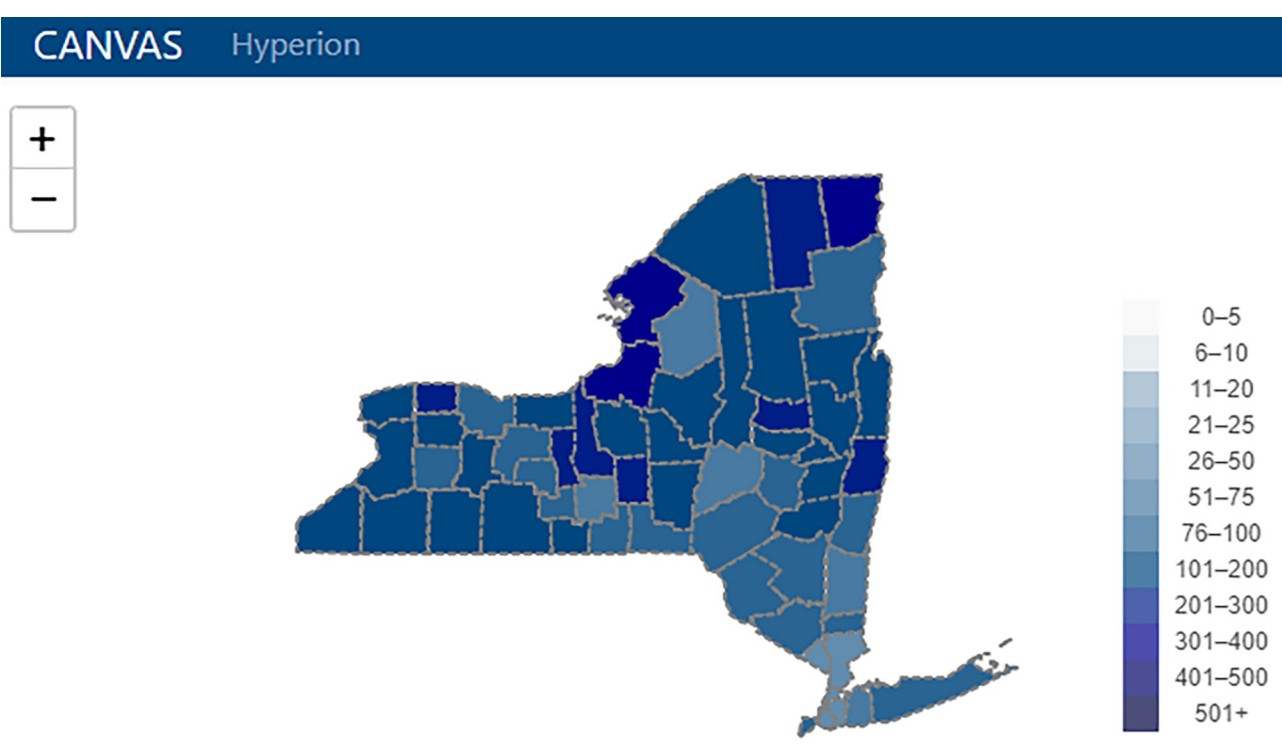

**Fig 3. CANcer Visual Analytic System (CANVAS) visual.** Map created using Leaflet with data from OpenStreetMap contributors.

insertion. Some validation routines are specifically applied to certain data sources. For example, correct address information is critical for analyses or visualizations involving geospatial elements (Fig 3). Incoming address data are validated against a database of extant addresses, and a "certainty percentage" is calculated which permits a user-defined accuracy threshold for certifying the address as correct. Validation routines are linked to a wizard and programmers are not needed for review of their output. The wizard allows users to examine items or processes flagged as potentially incorrect by the validation routines, and it presents them with a point-and-click menu of options (e.g., flag for further review, manually correct item, or approve upload).

To improve both computational and cost efficiency, HDM uses multi-threaded best practice approaches to parallelize processes and increase processing speed. Job duration calculations allow for efficient scheduling of multiple processor cores prior to task execution. Utilizing the combination of scheduling plus multi-threaded processing increased speed by 50–77% compared to non-optimized methods. Cost efficiency is further optimized using open-source technology (e.g., Python/Flask) for all code. Hyperion imports millions of rows of data every few hours with a total implementation cost of $1,500 for hardware and software (in addition to already existing enterprise licensure costs). This design also facilitates future implementation of analytic pipelines, including machine learning pipelines.

HDM sandboxes offer researchers temporary, partitioned access to curated datasets. Researchers can autonomously handle and analyze their data within a secure environment. Access is completely separated from other Hyperion architecture, and access privileges have an associated, pre-specified timeframe. Upon lapse of access privileges, HDM will terminate access and archive and lock the data.

## Hyperion front end

The front-end platform also utilizes open-source programming technology (HTML5 and Java-Script) to limit cost and align with common programming skillsets. Via a secure web portal, each user has access to a curated set of dashboards and activities based upon their access profile. Dashboards are highly interactive and customized to user groups based on iterative feedback. Users can view and securely sign documents (such as the data governance policy) and submit requests and ideas via an integrated ticketing system.

Hyperion administration and security activities are accessible via the web portal and HTML5 interface for approved users; point-and-click tools enable addition of new users and assignment of access privileges. The system has been tested and functions with all major web browsers. System administrators can view applications utilization data, including granular data by user and by clicks. Real-time data monitoring in conjunction allows for all data use to be precisely tracked and audited.

Hyperion's analytics-processing framework enables real-time analytics across any data element in the system, including those accessed via Application Programming Interfaces (APIs, e.g., imaging, pathology, and molecular/genomic data). Process efficiencies as described above permit rapid turnaround of ad-hoc reports, and scheduled reports are automatically generated and delivered at set intervals. A key principle guiding design of the user experience, however, prioritizes user autonomy in data access, analysis, and visualization. To this end, multiple customized dashboards permit users to directly interact with curated datasets and visualizations; Table 3 gives examples of some dashboards developed for various user groups in WCI.

Cross-referencing and validation of patient addresses (described above) as well as integrated data crosswalks between various geographic levels (e.g., zip code and census tract levels) facilitate geospatial visualization and analysis at any geographic level specified by the user. These capabilities enhance the ability to examine data stratified by area within the WCI catchment area, map changes over time, or link to other datasets to analyze disparities across the region (e.g., nutritional access or socioeconomic disadvantage. Hyperion's CANcer Visual Analytic System (CANVAS) allows users to create regional map overlays for data elements such as patient visits, diagnostic categories, and clinical trial accruals which can be toggled on/off or superimposed (Fig 3). Animations permit direct visualization of changes over time.

## Data governance

Data governance processes follow a centralized data governance model with a decentralized execution standard: this allows centralized control and authority to reside with the informatics team but permits the individual users and groups to be able to execute queries and analytics on curated sandboxed data sets or dashboards. This method of governance ensures data reliability and consistency via a single data baseline, validated daily.

Upon initial login, Hyperion users are presented with the current WCI data governance policy, which they must electronically read and sign prior to interacting further with the system (Fig 4). The integrated governance platform requires new signatures at login when the policy is updated. All signatures are encrypted and stored with the documents in Hyperion. The platform can support multiple documents with multiple versions, allowing for customized documentation for different user types as required.

Hyperion has an integrated custom ticketing platform for users to submit requests for additional development, reporting, or other requests. This is the primary method for initial communication of needs to the WCI Informatics Team and the Data Governance Committee. This method limits user cognitive load, streamlines development process, and facilitates automated tracking and reporting of requests. The ticketing platform collects all relevant info from users

**Table 3. Example user dashboards.** See supplementary material for sample visualizations.

| Dashboard | Content | Users |
|---|---|---|
| *Physician* | - Individual physician panel data<br> ○ Demographics<br> ○ RVU metrics<br> ○ Referral patterns<br> ○ Common medications prescribed<br> ○ Clinical trial accruals | - Physicians (access to own data only) |
| *Clinical Trials* | - Clinical trial accrual metrics, in total, by disease groups, and by trial<br>- Color coding for quick visual inspection of trials above or below accrual targets | - Clinical Trials Office<br>- Investigators |
| *Nursing* | - Patient and appointment numbers (clinic + infusion):<br> ○ Total<br> ○ By location<br> ○ By day of week and hour of day<br> ○ By regimen, with acuity score<br>- Appointment time metrics<br> ○ Average time per appointment<br> ○ Percent on-time appointments<br> ○ Length of appointment by hour of day and day of week<br>- Patient demographics<br>- Numerous point-and-click data filters available | - Nurses<br>- Nursing managers<br>- Infusion center staff<br>- Clinical operations staff |
| *Shared Resources* | - Usage tracking (rates and hours) for all WCI shared resources (e.g. biostatistics, genomics)<br>- Resource membership list tracking | - Shared resource leadership<br>- Administrators |
| *Pharmacy* | - Tracks timing of antineoplastic therapy order process steps and lab draws<br>- Visual displays, by provider, of where process may be impacting therapy delivery time | - Pharmacists and pharmacy staff |
| *CANVAS* | - Interactive map of catchment area, with configurable map overlays and animations using point-and-click functionality | - All Hyperion users |
| *Quality* | - Mortality rates with demographic breakdown<br>- Visual and numeric tracking of hospital admissions and antineoplastic therapy administration within 14 and 30 days of death overall and by disease grouping<br>- Readmission data<br>- Clinical decision tracking (therapy on-pathway, off-pathway, clinical trial)<br>- Point-and-click filter functionality | - Leadership<br>- Quality officers |

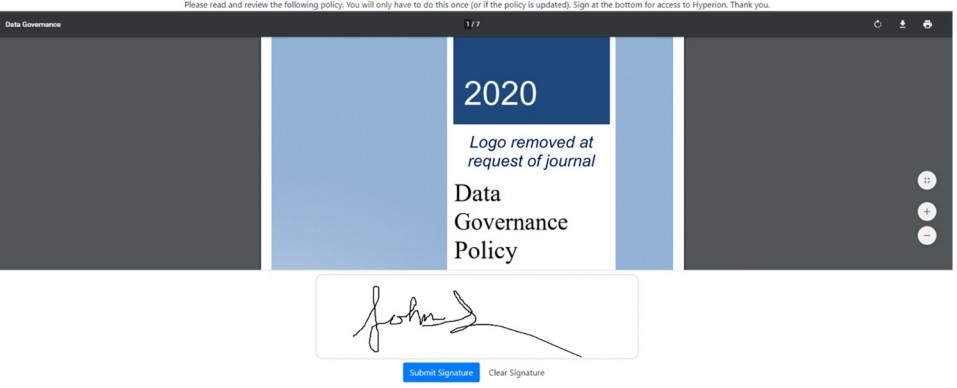

**Fig 4. Custom electronic signature platform presenting the data governance policy at login.**

and populates a document for committee discussion, triage, and project management. Tickets are automatically routed to individuals on the governing committee for initial approval; if more information is needed, the committee member can submit questions back to the system via embedded email links. For items with limited time and scope requirements, the team can independently triage the request and place it appropriately within the project queue. Larger requests (e.g., >40 person-hours, complexity, and/or multiple stakeholder groups affected) are flagged for committee discussion. Users can track the progress and actions on their individual tickets, including the ability to view key details entered by the informatics team (such as committee approvals, relevant dates, and requests for further information).

The Data Governance Committee consists of the WCI Informatics team and representatives from clinical operations, quality, administration, and research, all of whom have separate leadership roles within their respective domains. The Committee meets monthly and determines overall mission and priorities, discusses and triages large project requests, and reviews data usage and security.

## Security

Hyperion is only accessible via a secure networked computer or institutional virtual private network (VPN). Hyperion's custom security module augments the institution's Active Directory (AD) authentication and allows for user auditing and access control down to the individual application level. This permits user access to be customized by role and job function. and allows for the capture of full details on each user action, which Hyperion logs and monitors. Secure sandboxes are provided to users for data analytics purposes, without the ability to download data; this method of viewing and analyzing Hyperion reports is strongly encouraged. Otherwise, requested data documents are sent via either end-to-end encrypted email or secure file transfer protocol (SFTP). All documents able to be downloaded by the user include meta tags to identify the source of download, the individual user, and date/time stamps. In the event of a data consistency issue, the document can be compared to the logged SQL audit calls stored within Hyperion to ensure data was not altered. Request fulfillment for patient-identifiable data meets all institutional data security policies, including review by the Data Governance Committee.

## Usage metrics

From January 2020 through December 2021, Hyperion has processed >41 billion records. More than 174 million records are currently stored, and 791 million records have been updated since January 2020. Hyperion currently has 148 unique users (52% physicians, 21% nursing, 17% IT, 10% administrators) accessing an average of 27 pages per day through December 2021. Table 4 shows the most accessed dashboards. There are currently 25 customized real-time updating dashboards in Hyperion, as well as 13 automated reports that distribute throughout the week to various departments and individuals via automated email

**Table 4. Most accessed dashboards, January 2020 –December 2021.**

| Application Name | Total Visits |
|---|---|
| Clinical Trials Dashboard | 4690 |
| Physician Dashboard | 2821 |
| CANVAS | 1604 |
| Nursing Dashboard | 1586 |
| Shared Resources | 1098 |
| Referrals | 601 |

distribution. The average turnaround time for an ad-hoc reporting request is 3 days. The average time to deploy a new dashboard is about 4 weeks.

## Discussion

Creating and maintaining complex, secure, and high-volume data warehouses, processing and assembling data views, and interfacing new data sources represent significant challenges for academic healthcare organizations, even those with adequate information technology (IT) staffing and budget. In typical practice, the team (or vendor) that creates the data warehousing software is distinct from the team tasked with maintaining it and supporting data access and visualization activities. Even after deployment of the data warehousing software, basic IT support for data warehousing typically includes interface developers, application developers, business analysts, Business Intelligence developers, security professionals, and help desk staff. Iterative adaptation of the software to meet changing data needs can be challenging or impossible, leading to the accrual of "workarounds" and technical debt.

In WCI, we assembled a small, transdisciplinary team to develop a customized, adaptable, and scalable data management approach, supported by senior leadership and enterprise IT structures. Beginning with the definition and elaboration of key challenges we wished to address, we designed and built a modular and scalable software package addressing data storage, validation, and processing needs as well as data monitoring, access, and visualization. These structures were designed to permit rapid iteration and adaptation by a small but highly skilled technical team when needed, but to allow basic administration and continued maintenance to be performed by non-technical staff. The Hyperion database architecture, Hyperion Data Manager, security module, and governance modules were designed and deployed with a 6-month timeline, and the entire package can be maintained by a single full-time equivalent (FTE). The modular extensible approach significantly reduces enhancement and update times. Limiting the continuous need for high-level technical skillsets to maintain software and data integrity frees a smaller team to work "at the top of their cognitive skillsets", iterating solutions in response to user feedback, and creatively generating new solutions to complex data problems in WCI. This approach maximizes cost-effectiveness in addition to overall efficiency. Furthermore, Hyperion aligns with FAIR data principles (findability, accessibility, interoperability, and reusability). [12]

Hyperion has high user uptake, with many faculty members, staff, administrators, and others logging in daily to independently access data for clinical, operational, administrative, and research purposes. Although much of the data in Hyperion originates within the institutional EHR, it has previously been sequestered within individual patient records and only slightly more accessible to automated extraction methods than paper charts. Hyperion makes validated, curated, organized, near-real-time datasets automatically accessible and easily explorable via an interactive suite of analysis and visualization tools. Quality initiatives, program development, physician decision-making, clinical trial planning and management, research (grant applications and peer-reviewed analyses), clinical operations management, and more are all supported within a single management platform.

In addition to design and build of the software elements, implementation of Hyperion required careful consideration and design of support structures to address data governance, security, and project management. Intrinsic software elements such as the data governance policy tracking, security audits, and the sophisticated ticketing system are embedded in a set of processes overseen by a supporting committee structure consisting of key stakeholders and meeting at least monthly. Users are not competing with requests external to WCI for attention, and two-way communication is streamlined and efficient due to the embedded nature of the informatics team.

Beyond embedding a responsive team within and among the software users, the composition and functioning of the team are substantial revisions to the usual model of academic IT teams. A co-directed model shares formal authority for external interactions between a technical expert with extensive background in healthcare IT, and a practicing clinician with training in data science and informatics. Internally, the team is structured with a flattened decisional hierarchy, a deliberate emphasis on diversity of opinion and perspective, and a rapidly iterative approach to problem solving adapted from industry. Projects are managed through a combination of agile-based approaches and more traditional project management philosophies such as milestone phase gating. Although this structure is not able to completely shield the team from institutional bureaucracy, it has helped to create and sustain space for innovation and creativity within a traditionally cautious and even inert system.

There are several potential disadvantages to our strategy for addressing data needs within an academic cancer center. Academic medical centers may not be able or willing to support in-house software development for patient data, relying on outsourced software to guarantee robustness, security, and technical support. In the current market, the technical skillsets to achieve customized software solutions for managing patient data are rare and expensive, and achieving buy-in to budget for these positions may be very challenging. We have mitigated personnel costs with a smaller team comprised of individuals with cross-functional skillsets, but this poses its own difficulty: there is limited redundancy within the system to accommodate team member absences or departures. Although the software design offloads much of the technical maintenance, allowing it to be done by nontechnical personnel, the sustainability of the software still relies on a core highly-skilled technical team. However, stable teams are also required to internally support many vendor products within the technical ecosystem.

In summary, Hyperion has surmounted large challenges in working with healthcare data to merge, organize, validate, and package data for use in multiple applications including interactive user dashboards. Additional design considerations included lowering the skill floor for interaction with and maintenance of the software, reducing costs, and encouraging user autonomy. Development of data governance and other support structures, as well as discussions about team functioning and structure, occurred in parallel with the software build. Future work includes turning our attention to further supporting data analytics, including machine learning (ML). A ML pipeline is being developed to allow users to explore their data using advanced techniques while also receiving hands-on education about the potential and pitfalls of these methods. Novel data visualization methods, including augmented and virtual reality methods (AR/VR) are also in initial development and testing. Hyperion provides a flexible, reliable, and scalable foundation for the development of responsive and innovative applications to support the mission and goals of an academic cancer center.

## Supporting information

**S1 Text. Fig A**. Nursing dashboard. **Fig B.** Example of Provider Dashboard landing page, with interactive features (hover-over pop-up text and ability to click on bar graphs to "drill down" on data). **Fig C.** Clinical Trial dashboard main page, with interactive features. (DOCX)

## Acknowledgments

We acknowledge the continued support of the Wilmot Cancer Institute's leadership in encouraging innovation, as well as the support of the physicians, nurses, and staff.

## Author Contributions

**Conceptualization:** Eric Snyder.

**Data curation:** Eric Snyder, Thomas Rivers, Lisa Smith, Scott Cunliffe.

**Funding acquisition:** Erika Ramsdale.

**Methodology:** Eric Snyder, Thomas Rivers, Lisa Smith, Scott Paoni, Scott Cunliffe, Arpan Patel, Erika Ramsdale.

**Project administration:** Eric Snyder, Thomas Rivers, Lisa Smith, Scott Paoni, Scott Cunliffe.

**Resources:** Eric Snyder.

**Software:** Eric Snyder, Thomas Rivers, Lisa Smith, Scott Paoni, Scott Cunliffe, Erika Ramsdale.

**Supervision:** Eric Snyder.

**Validation:** Eric Snyder, Thomas Rivers, Lisa Smith, Scott Paoni, Scott Cunliffe.

**Visualization:** Eric Snyder, Lisa Smith, Scott Cunliffe.

**Writing – original draft:** Eric Snyder, Erika Ramsdale.

**Writing – review & editing:** Eric Snyder, Thomas Rivers, Lisa Smith, Scott Paoni, Scott Cunliffe, Arpan Patel, Erika Ramsdale.

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
