## [Decision Letter · Decision Letter 0]

15 Jun 2022

PDIG-D-22-00095

From months to minutes: creating Hyperion, a novel data management system expediting data insights for oncology research and patient care

PLOS Digital Health

Dear Dr. Ramsdale,

Thank you for submitting your manuscript to PLOS Digital Health. Based on reviewer comments and questions, we feel that it has merit but does not fully meet PLOS Digital Health's publication criteria as it currently stands. Therefore, we invite you to submit a revised version of the manuscript that addresses the points raised during the review process.

We look forward to receiving your revised manuscript.

Kind regards,

Rutwik Shah, MD

Guest Editor

PLOS Digital Health

Journal Requirements:

2. Please ensure that the funders and grant numbers match between the Financial Disclosure field and the Funding Information tab in your submission form. Note that the funders must be provided in the same order in both places as well.

3. Please update your Competing Interests statement. If you have no competing interests to declare, please state: “The authors have declared that no competing interests exist.”

5. Please provide separate figure files in .tif or .eps format and ensure that all files are under our size limit of 10MB. 

For more information about how to convert your figure files please see our guidelines: 

6. All figures and supporting information files will be published under the Creative Commons Attribution License (creativecommons.org/licenses/by/4.0/). Authors retain ownership of the copyright for their article and are responsible for third-party content used in the article. 

Figure 3: please (a) provide a direct link to the base layer of the map used and ensure this is also included in the figure legend; (b) provide a link to the terms of use / license information for the base layer. We cannot publish proprietary or copyrighted maps (e.g. Google Maps, Mapquest) and the terms of use for your map base layer must be compatible with our CC-BY 4.0 license. 

Please upload any written confirmation as an 'Other' file type. It must clarify that the copyright holder understands and agrees to the terms of the CC BY 4.0 license; general permission forms that do not specify permission to publish under the CC BY 4.0 will not be accepted. Note that uploading an email confirmation is acceptable.

Additional Editor Comments (if provided):

Reviewers' comments:

Reviewer's Responses to Questions

**Comments to the Author**

1. Does this manuscript meet PLOS Digital Health’s publication criteria? Is the manuscript technically sound, and do the data support the conclusions? The manuscript must describe methodologically and ethically rigorous research with conclusions that are appropriately drawn based on the data presented.

Reviewer #1: No

Reviewer #2: Yes

Reviewer #3: Yes

2. Has the statistical analysis been performed appropriately and rigorously?

Reviewer #1: N/A

Reviewer #2: N/A

Reviewer #3: N/A

3. Have the authors made all data underlying the findings in their manuscript fully available (please refer to the Data Availability Statement at the start of the manuscript PDF file)?

Reviewer #1: Yes

Reviewer #2: No

Reviewer #3: Yes

4. Is the manuscript presented in an intelligible fashion and written in standard English?

Reviewer #1: Yes

Reviewer #2: Yes

Reviewer #3: Yes

5. Review Comments to the Author

Reviewer #1: General comments

This paper highlights the design and implementation of an in-house customized data management software in an academic medical center involving a small, cross-functional team. This is an excellent practical project and seems to work well for a specific context. However, for digital health and health informatics, the methods and measures are unclear and don’t have adequate rigor. I am also unsure about the generalizability of the study findings 

Specific comments

Major comments

1. Some of the challenges that have been identified are very specific to a context. It would be helpful to address the generalizability of these challenges to diverse cancer settings (academic and community centers). Also, what methods were used to determine these challenges? How did the authors ascertain that these were the “only” challenges?

2. Also, how were the design elements for each challenge developed. Replicability of these findings is difficult, and this study seems more like a solution to a contextual problem. 

3. It is also clear if any user feedback on ease of use was measured. Also, were the end-users involved in design and development of Hyperion? Please include more details. The authors have included usage metrics and these metrics would often cloud the overall usability and end-user preferences. 

4. The authors make a case for a successful implementation of custom data management software but haven’t measured any implementation outcomes. 

5. It is not clear what were the novel design ideas in data architecture. 

6. It would be helpful to also provide a rationale on why an in-house customized data management software was required and provide some generalizable knowledge on how academic medical centers should take decisions for or against the development of in-house customized data management software. Much of the limitation section highlights all the issues on why academic medical centers are unwilling to support in-house software development for patient data and it could be that this is a “unique” situation. The generalizability of this study findings is what I am struggling most. This is an excellent practical project with great value to a specific context. However, it lacks rigor in methods, measures, and generalizability 

Minor Comments

1. Figure 5 is titled most accessed pages by end users through October 2021. Instead of merely presenting the pages from 1-6 ( clinical trial dashboard- Referrals), it would be helpful if the number of times each of these pages have been accessed per a specific time period or the average number for a defined time period.

Reviewer #2: This paper describes the design and implementation of a data management platform denominated Hyperion for an academic cancer center. The authors present all the features and benefits of using this interface in their center. The authors discuss the limitations of Hyperion in the discussion section and future work which looks promising. 

I only have minor comments bellow.

References could be added to several statements in the manuscript. Regarding related work, the authors only mention one institution (reference 4). This could be extended to include more examples since the authors refer to “Some institutions”.

It would be interesting to visualize graphical examples of tasks or dashboards which are possible with Hyperion as described on the paper, as supplementary material.

Please check the references in the manuscript and include those currently after the dot before the final dot, examples in “while adhering to already thin financial budgets.[2]” and “skilled personnel.[5]”.

The first sentence in Data Governance section should be divided into at least 2 sentences.

In Usage metrics specify the start and end dates, instead of ‘through October 2021’, so that the reader understands the referred time period. Figure 5 legend should also include this information.

In Figure 1 please add the acronym meaning for your center WCI.

Figure 2 is missing y and x axis legends.

Reviewer #3: Hyperion seems to be a very useful system, bringing together data within a single cancer center. However the paper should be improved significantly. I have a number of suggestions:

Please follow the submission guidelines of the journal (https://journals.plos.org/digitalhealth/s/submission-guidelines):

- Abstract is conceptually divided into three sections (Background, Methodology/Principal Findings, and Conclusions/Significance), do not apply these distinct headings to the Abstract within the article file.

- Include an 150-200 word author summary

- Word count: according to the submission guidelines, there is no limit of 4000 as mentioned in your paper

The background information is quite limited. What are state-of-the-art Clinical Data Management (CDM) systems? How does Hyperion compare to these? It might be good to include a table with features, comparing current CDM systems.

The number of references (7) is very small. Sites/software such as REDCap, OnCore, clinicaltrials.gov should be referenced, using a scientific publication (e.g. REDCap: https://www.sciencedirect.com/science/article/pii/S1532046408001226).

It's surprising that a paper about a clinical data management platform does not discuss the FAIR Guiding Principles (https://www.nature.com/articles/sdata201618). How does Hyperion handle these four aspects?

The Hyperion front-end works with HTML5 and JavaScript. Was it tested in all major browsers (Chrome/Firefox/Edge/Safari/Opera)?

Probably the most difficult task in a custom-made CDM system is security. I miss the details on security in this paper. For example, does Hyperion support two-factor authentication?

Is there a way to access imaging data (e.g. DICOM) through Hyperion, for example in the Physician Dashbord? The same for genomics data, digital pathology data, etc.: does Hyperion contain a link to these raw data?

A screenshot of one of the other dashboards than just CANVAS would be useful, for example the Clinical Trials Dashbord or Physician Dashboard (dummy data could be used), since these are most accessed?

6. PLOS authors have the option to publish the peer review history of their article (what does this mean?). If published, this will include your full peer review and any attached files.

**Do you want your identity to be public for this peer review?** For information about this choice, including consent withdrawal, please see our Privacy Policy.

Reviewer #1: No

Reviewer #2: Yes: Marta Fernandes

Reviewer #3: Yes: Tim Hulsen

---

## [Editor Report · Decision Letter 1]

4 Sep 2022

From months to minutes: creating Hyperion, a novel data management system expediting data insights for oncology research and patient care

PDIG-D-22-00095R1

Dear Dr. Ramsdale,

We are pleased to inform you that your manuscript 'From months to minutes: creating Hyperion, a novel data management system expediting data insights for oncology research and patient care' has been provisionally accepted for publication in PLOS Digital Health.

Best regards,

Rutwik Shah, MD

Guest Editor

PLOS Digital Health